# Spatial and temporal correlations in human cortex are inherently linked and predicted by functional hierarchy, vigilance state as well as antiepileptic drug load

**Paul Manuel Müller** [1,2,3], **Christian Meisel** [1,2,3,4,5]*

**1** Computational Neurology Lab, Department of Neurology, Charité–Universitätsmedizin Berlin, Berlin, Germany, **2** Berlin Institute of Health, Berlin, Germany, **3** NeuroCure Cluster of Excellence, Charité–Universitätsmedizin Berlin, Berlin, Germany, **4** Center for Stroke Research Berlin, Berlin, Germany, **5** Bernstein Center for Computational Neuroscience, Berlin, Germany

* christian@meisel.de

**Data Availability Statement:** All data is available within the Epilepsiae data repository (http://epilepsy-database.eu). The code to analysis is

## Abstract

The ability of neural circuits to integrate information over time and across different cortical areas is believed an essential ingredient for information processing in the brain. Temporal and spatial correlations in cortex dynamics have independently been shown to capture these integration properties in task-dependent ways. A fundamental question remains if temporal and spatial integration properties are linked and what internal and external factors shape these correlations. Previous research on spatio-temporal correlations has been limited in duration and coverage, thus providing only an incomplete picture of their interdependence and variability. Here, we use long-term invasive EEG data to comprehensively map temporal and spatial correlations according to cortical topography, vigilance state and drug dependence over extended periods of time. We show that temporal and spatial correlations in cortical networks are intimately linked, decline under antiepileptic drug action, and break down during slow-wave sleep. Further, we report temporal correlations in human electrophysiology signals to increase with the functional hierarchy in cortex. Systematic investigation of a neural network model suggests that these dynamical features may arise when dynamics are poised near a critical point. Our results provide mechanistic and functional links between specific measurable changes in the network dynamics relevant for characterizing the brain's changing information processing capabilities.

## Author summary

A growing body of research suggests spatial and temporal correlations, which capture the propagation of information in space and time, to be useful characterizations of information processing in the human brain. The criticality hypothesis, the hypothesis that networks in the brain reside in the vicinity of a phase transition, posits that spatial and temporal correlations are intimately linked and maximized near the critical point.

available under https://gitlab.com/computational-neurologie/iEEG_STC.git.

**Funding:** PM and CM are funded by Charité — Universitätsmedizin Berlin. CM and PM are supported by NeuroCure Cluster of Excellence, funded by the Deutsche Forschungsgemeinschaft (DFG, German Research Foundation) under Germany´s Excellence Strategy EXC 2049-89829218. CM acknowledges support from a NARSAD Young Investigator Grant. The funders had no role in study design, data collection and analysis, decision to publish, or preparation of the manuscript.

**Competing interests:** The authors have declared that no competing interests exist.

Previous research has predominantly focused on spatial and temporal correlations independently and was often restricted in duration, thus limiting our knowledge whether spatial and temporal correlations indeed co-vary and what other factors influence these information integration properties in general. Here, we use long-term invasive EEG data to comprehensively map temporal and spatial correlations according to cortical topography, vigilance state, and drug dependence over extended periods of time. We show that temporal and spatial correlations in cortical networks are strongly linked, decline under antiepileptic drug action, and completely break down during slow-wave sleep. We provide direct electrophysical evidence that temporal correlations follow a gradient which aligns with the functional hierarchy. Systematic investigation alongside a companion neural network model suggests that these findings may arise due to dynamics being poised near a critical point.

## Introduction

An essential ingredient for information processing is thought to be the ability of neural circuits to integrate information over appropriate periods of time and across different cortical areas. Characterization of neural systems according to their temporal and spatial correlations (TC and SC, respectively) has consequently provided important insight into their information processing capabilities. For example, in decision-making and working memory tasks [1–4], the ability to integrate over extended periods of time may increase the signal-to-noise ratio and afford to maintain some memory of past activity. In non-human primates, temporal correlations have been found to increase along the functional hierarchy providing a unifying principle for information integration across different timescales [1,5,6]. Similarly, the ability to integrate information in space, across functionally specialized regions, is considered essential for normal brain functioning [7–9]. Consequently, effective propagation and integration of information in space has been shown to depend on vigilance state where it is maximized during wake and breaks down during slow-wave sleep [10].

While temporal and spatial correlations are thus promising in providing a principled approach to characterizing information integration, essential questions related to them remain only incompletely understood. First, temporal and spatial correlations have mostly been studied independently. An understanding if and how temporal and spatial correlations are related is still missing. Second, previous studies have largely been limited in duration, focusing on certain tasks or windows of interest, thus providing only an incomplete picture of the variability and fluctuations of these indices, their potential dependence on vigilance state and interventions, such as medication treatment. Third, it is still unclear whether concepts like the hierarchical ordering of temporal scales also apply to human brains. Although there is strong evidence for a hierarchical ordering of temporal correlations in non-human primates [6], evidence in humans has been limited to comparisons between core and periphery networks in MEG [11] or fMRI [12]. Thus, whether the concept of hierarchical ordering of temporal correlations, as measurable in iEEG, along processing pathways similarly applies to human brains remains currently unresolved [13].

Here we use long-term invasive EEG data to comprehensively map temporal and spatial correlations according to cortical topography, vigilance state and drug dependence over extended periods of time. We show that temporal and spatial correlations in cortical networks are intimately linked, break down during slow-wave sleep (SWS) and decline under antiepileptic drug (AED) application. Further, we report that temporal correlations in human

electrophysical signals follow a gradient which aligns with the functional hierarchy. Finally, we study a neural network model that reproduces the tight interconnection between spatial to temporal correlations, their hierarchical ordering, and decline under AED action. Our results provide novel mechanistic and functional links between specific measurable changes in the network dynamics relevant for characterizing the brain's changing information processing capabilities.

## Results

We analysed invasive electroencephalography (iEEG) recordings from 23 patients to characterize spatial and temporal correlations (STC) as functions of vigilance state, antiepileptic drug (AED) action, and of functional cortical hierarchy. Broadband $\gamma$-power (56–96 Hz) was used as an index of population firing rate near an electrode [14–18] to derive spatial and temporal correlations and based on prior comparative work across broad frequency ranges. In line with these prior studies, we observed this high-frequency domain activity to be best suited to resolve the correlations within cortical dynamics compared to other, lower frequency bands (see S2 and S3 Figs for TC and SC respectively). Specifically, we measured the decay speed of the autocorrelation function of this signal as an index for temporal correlations (TC; [5,19]) and the decay of the pairwise cross-correlation function over distance as an index of spatial correlations (SC; [20–22]; Fig 1).

### Spatial and temporal correlations are tightly interconnected

Previous research on STC as a metric for information integration in cortex has primarily focused on task-related activity or investigation of relatively short periods, thus providing only

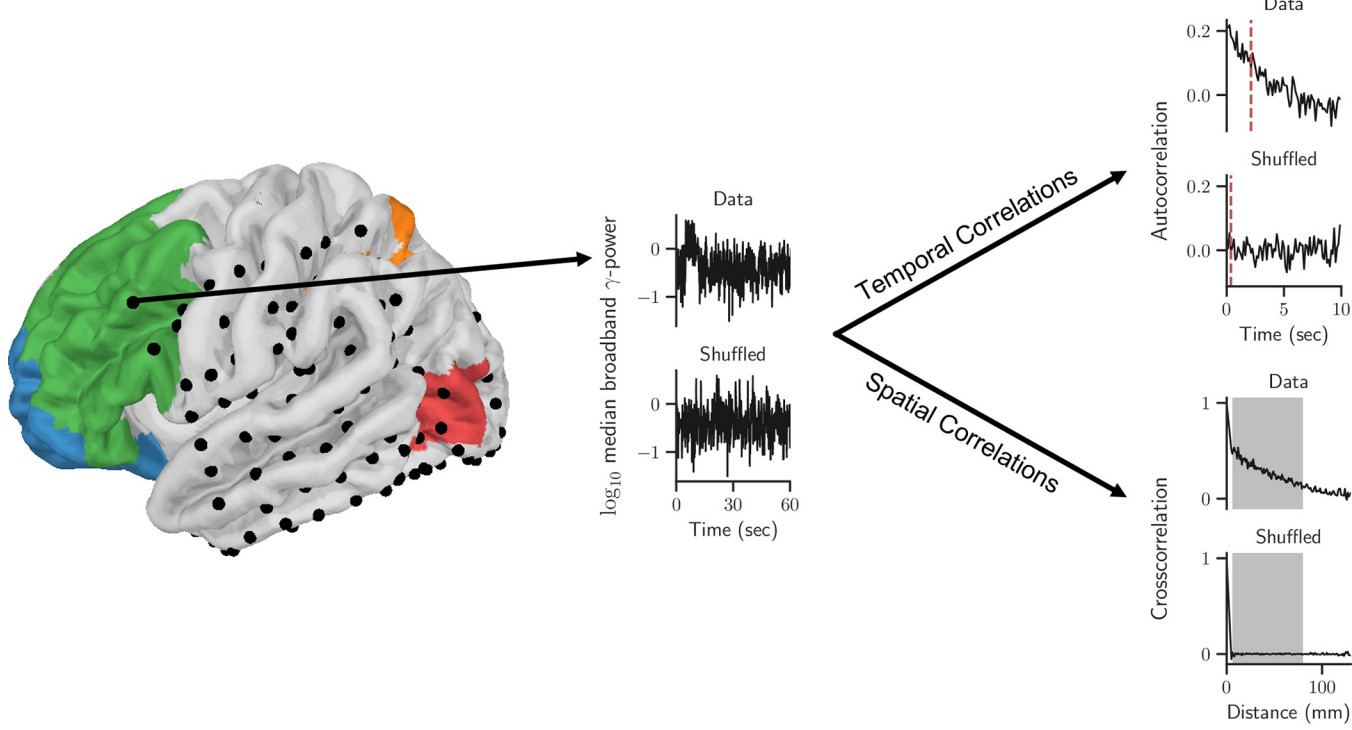

**Fig 1. Topographical mapping of temporal and spatial correlations in cortex over extended periods of time. Left:** Projection of invasive EEG electrodes from an exemplary subject with brain surfaces along the cortical hierarchy colour coded. **Middle:** γ-power timeseries for original and time-shuffled data. **Right:** Autocorrelation functions (top two panels) and cross-correlation functions (bottom two panels) are used to calculate temporal and spatial correlations, respectively.

limited insight into fluctuations of STC over time, their co-variation, and dependence on external factors, such as medications. We observed TC and SC to fluctuate significantly over the course of a full day (Fig 2A and 2B). Importantly, SC and TC were highly correlated, i.e., when TC increased, SC tended to increase as well and vice versa. Fig 2C shows this strong co-variation for one patient (individual results from all patients can be found in S6, S7, and S8 Figs). Notably, this co-variation was destroyed in time-shuffled surrogate data with the same $\gamma$-power (Fig 2D). A highly significant correlation was observed in all but two patients (average Spearman rank correlation $\rho = 0.5$, Fisher combined $p$ value below machine precision).

## Spatio-temporal correlations break down during slow-wave sleep

Next, we investigated whether fluctuations of STC over the course of the day (midnight to midnight; Fig 2A and 2B) depended on vigilance state. To investigate the dependence of STC on vigilance states, we used a validated algorithm to identify periods of slow-wave sleep (SWS; [23]). TC and SC were decreased during periods of SWS (Fig 2A and 2B; green bars). Across patients on low AED days, the reduction in TC and SC during SWS periods as compared to the remaining time periods (nonSWS) was significant ($p<0.001$ and $p<0.05$ for TC and SC, respectively; Wilcoxon signed-rank test; Fig 2G and 2H). No changes were observed for the time-shuffled surrogate data (Fig 2G and 2H, grey bars). TC were significantly reduced during SWS irrespective of the AED load (S2 Fig panel E). The decrease of SC during SWS was not significant anymore when looking at times other than low AED days (S3 Fig panel E).

## Spatio-temporal correlations decline under antiepileptic drug action

Beyond the fluctuations over the course of a day, we observed TC and SC levels to depend on AED load (Fig 2A and 2B). We compared AED dosages based on the summed prescribed dosage normalized by the defined daily dosage and did not differentiate between different AED due to the high variability of AED given across patients (see S2 Table). Even though, different AED have different mechanisms of action, including ion-channel blockers, e.g., Phenytoin, GABAergic drugs, e.g., Clobazam, which increase inhibition, previous work has suggested that their influence on STC may be similar [19]. Comparison between full days with highest and lowest AED load in each patient indicated both TC and SC to be higher during the low AED days (Fig 2A and 2B, top panels). Fig 2E and 2F show the average autocorrelation and cross-correlation functions over all patients across low and high AED load days, respectively. Auto- and cross-correlation functions decayed slower for the low AED days (orange lines). Conversely, surrogate data (generated by shuffling the broadband $\gamma$-power time series for all segments; dotted lines in Fig 2E and 2F) exhibited an almost instantaneous decay indicating that changes were not a consequence of different power levels but reflective of the temporal and spatial correlation structure in the data. Consequently, TC and SC were significantly reduced for high AED load days during nonSWS episodes ($p<0.05$; Wilcoxon signed-rank test; Fig 2G and 2H). No changes depending on AED load were observed in time-shuffled surrogate data (grey bars in Fig 2G and 2H). No further decline of neither TC nor SC on high AED days was observed during SWS episodes (S2 Fig panel E and S3 Fig panel E). Only for TC a decline during high AED was observed when investigating the data irrespective of sleep stage (S2 Fig panel E).

## Temporal correlations increase with cortical functional hierarchy

Previous work in non-human primates has indicated that cortical areas exhibit a hierarchical ordering in their timescales of their temporal correlations [6]. The sparse spatial sampling with only a few electrodes per region only allowed the evaluation of TC but not SC as a function of

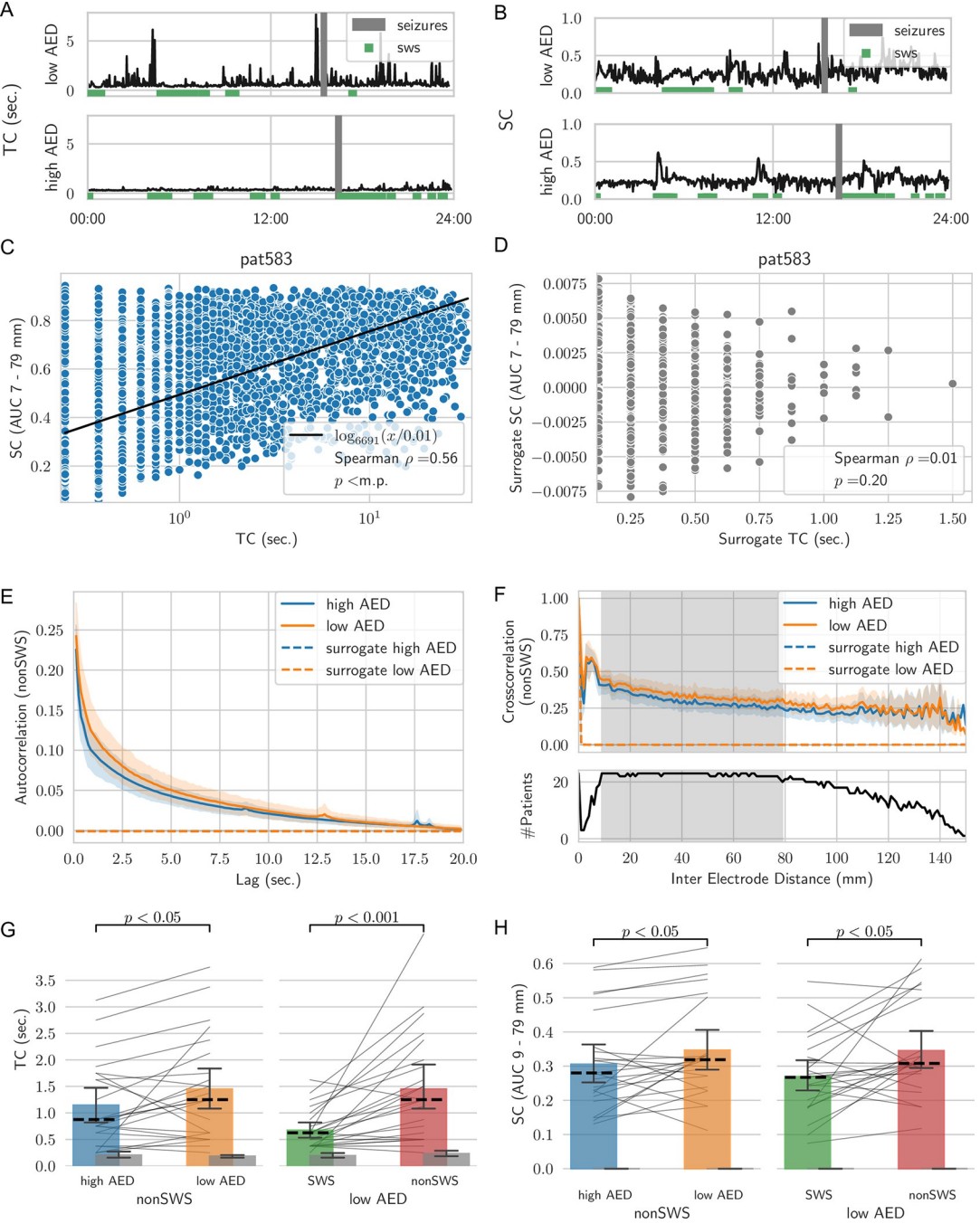

**Fig 2. Spatial and temporal correlations (SC, TC) are tightly linked, decline under AED action and break down during slow-wave sleep (SWS). A, B** Timeseries of TC and SC over one day for high and low AED dosages for one exemplary patient. SWS is marked with green squares and seizures (grey bars) are excluded from the analysis. **C** Co-variation of SC and TC shown for one patient. The black line indicates a logarithmic fit. **D** Surrogate data of SC and TC are overall smaller and exhibit no co-dependence. **E, F** Autocorrelation and cross-correlation functions averaged over all patients exhibit a faster decline during high AED load days. The grey shaded area in F was used for the SC calculation. **G, H** Reduction of TC and SC during SWS and high AED load (Wilcoxon signed rank test). Single patient values are indicated as thin lines, the median is given as dashed line and whiskers extend to the 95% confidence interval calculated via bootstrapping. Surrogate data (grey bars) exhibit no difference.

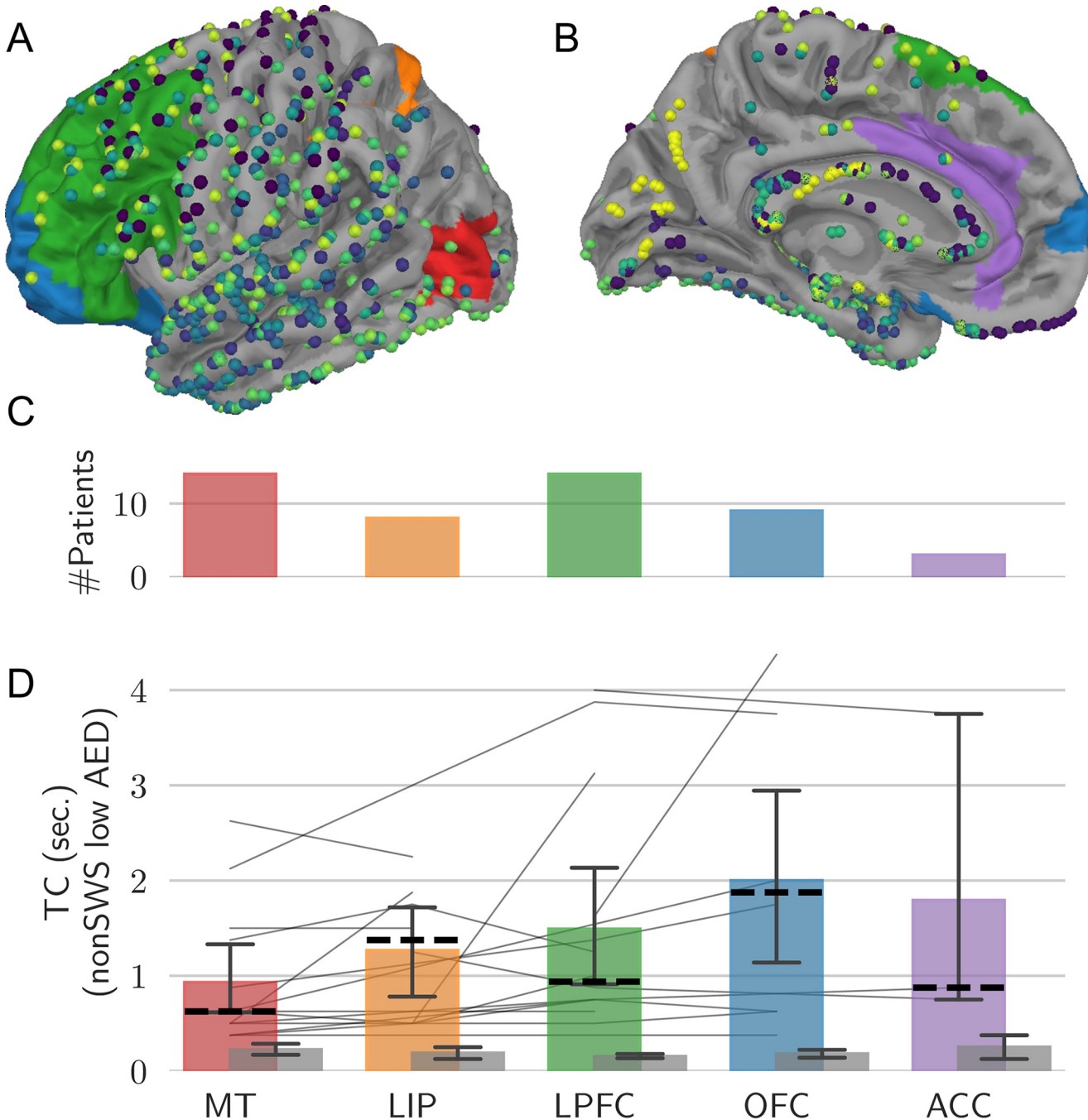

**Fig 3. Temporal correlations increase with functional hierarchy. A, B** Lateral and medial view of the left-hemisphere of a reference brain with electrodes of all patients projected onto the surface with coloured areas similar to [6]. **C, D** Number of electrodes and TC for each region. Thin lines indicate data from individual patients, coloured bars averages and dashed lines median values across patients. Whiskers denote the 95% confidence interval.

functional hierarchy in our data. To determine the existence of a hierarchical ordering of TC also in human cortex, we identified the five cortical regions along the visual pathway which are similar in function to the regions used for the analysis in non-human primates in [6]. The areas are colour-coded in Fig 3A and 3B, and electrodes of all patients are projected onto this reference cortical surface, collectively providing a dense sampling across cortex. As electrode

placement was solely determined by clinical considerations, each patient had electrodes in different positions and the number of patients per cortical region varied between 3 patients with electrodes in ACC to up to 14 patients with electrodes in MT or LPFC (Fig 3C). Fig 3D shows TC derived from the five cortical areas during low AED days and nonSWS. We focused this analysis primarily on low AED load and nonSWS episodes in an attempt to more closely resemble unperturbed, non-sleep related brain activity. TC tended to increase from regions with low hierarchy to regions higher in hierarchy. The average slope of increase was 0.4±0.2 sec./area ($p<0.05$; Wilcoxon signed-rank test). No increase with functional hierarchy was observed in the time-shuffled surrogate data (Fig 3D, grey bars).

## Dynamical network model reproduces co-variations of spatial and temporal correlations, their decline under antiepileptic drug action and hierarchical ordering

To gain insights into the mechanisms underlying the observed co-variation of spatial and temporal correlations, their hierarchical ordering and decline under AED action we compared our experimental findings with simulations of a parsimonious neural network [19,24,25]. The model or derivations thereof have been used in numerous studies due to it being sufficiently simple to provide insight into the mechanisms governing collective network dynamics yet entailing sufficient detail to model relevant aspects of regional variations in STC and AED action on network interactions. In short, the model consists of a 2-dimensional lattice of all-to-all connected excitatory and inhibitory neurons whose connection strengths decrease with increasing distance. During a simulation run, all connection strengths are kept constant as we want to explicitly investigate the effects of different connection strengths. The single neurons fire randomly with a certain probability as well as depending on their inputs. Excitatory synaptic inputs increase the likelihood of firing, inhibitory inputs decrease it. Even though self-connections are omitted in the model there are still loops via other neurons that may afford recurrent activity. Generally, network dynamics in this model are governed by the connection strengths which can be characterized by the largest eigenvalue $\lambda$ of the connectivity matrix [24,25]. Additionally, the neuron network model allows mimicking the effects of AED by either decreasing the excitability of neurons by $f_{exc}$, for example via ion channel blockers, or by increasing inhibitory synaptic strength by $f_{inh}$, for example via GABAergic drugs. As the results for both mechanisms are qualitatively similar in the model, we only report them for the former (for a direct comparison see also [19]).

In the absence of AED action, collective dynamics exhibited a phase transition from low activity to a high-activity phase when connection strength was increased. STC increased when $\lambda$ was increased to the critical value of $\lambda = 1$ (Fig 4E and 4F). As TC decreased again in the supercritical phase ($\lambda>1$, S9 Fig panel B), SC kept increasing further (S9 Fig panel A). This can be explained by SC being defined as the area under the cross-correlation function, which generally gets an offset to higher values for increasing $\lambda$. However, when SC were defined exactly as the TC, i.e., by quantification of the decay of the cross-correlation function, SC also peaked at $\lambda = 1$, as would be expected at criticality (S9 Fig panel C). Quantifying this decay rate, however, requires robust estimates of the correlation function at short distances which were not available in the data due to fixed interelectrode distances and low number of electrode pairs and patients in this range (Fig 2F). We thus show SC as measured by the area under the cross-correlation curve for comparability to the data. Thus, for interpretations with respect to criticality, consideration of both SC and TC collectively is preferable. This is supported by concomitant changes in the Hurst exponent, a commonly used measure to quantify the self-affinity of a time-series [26,27]. The Hurst exponent peaked at $\lambda = 1$, where it exhibited values well above

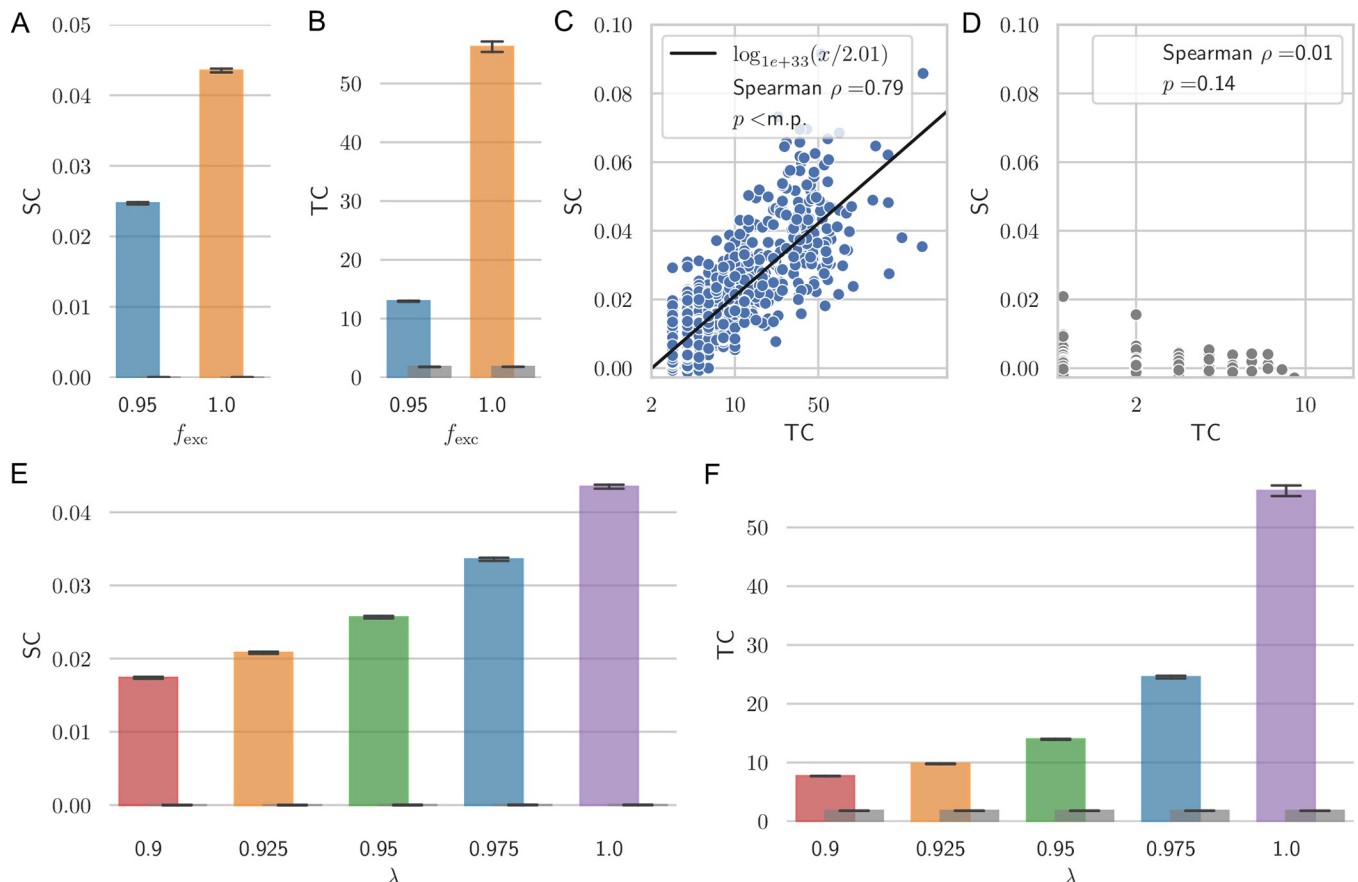

**Fig 4. STC in a neural network model.** Model simulations are shown for N = 1600 neurons on a 2-dimensional grid with periodic boundary conditions and distance-dependent connectivity strength. Other parameters are $f_{inh} = f_{exc} = \lambda = 1$, if not stated otherwise. One neuron was activated randomly every $max_t$ = 2000 time steps. Coloured bars give the ensemble average with whiskers extending to the 95% confidence interval. Results for surrogate data (shuffled time series) are given as grey bars. **A, B** The state of high AED can be mimicked in the model by decreasing the excitability of excitatory neurons, i.e., $f_{exc}$ = 1→0.95, leading to decreased SC, TC. **C** Co-variation between TC and SC for different model realizations with $\lambda \in$ {0.7, 0.75, 0.8, 0.85, 0.9, 0.95, 1.0} (10⁴ runs each, 7·10⁴ runs in total, Spearman ρ = 0.79, p<m.p. (machine precision)). The black line shows a logarithmic fit. **D** No correlations are found in time-shuffled data. **E, F** SC and TC increase with the absolute value of the largest eigenvalue λ approaching its critical value of λ = 1 similar to the hierarchical increase of TC in iEEG data (Fig 3).

0.5, which would be expected for uncorrelated signals (see S9 Fig panel D). Implementation of AED action decreased SC and TC (Fig 4A and 4B; here for a reduction from $f_{exc}$ = 1≙low AED load to $f_{exc}$ = 0.95≙high AED load). Similarly, a reduction of STC for smaller $f_{exc}$ could be observed for a broad range of λ (e.g., chosen from λ∈{0.9, 0.925, 0.95, 0.975, 1}, not shown) to account for the AED action on networks instantiated at different proximities to criticality. The reduction of STC under AED action in the model thus closely captured the decrease under higher AED loads in iEEG data (compare Fig 2E and 4F). In addition, SC and TC for different λ exhibited a robust covariation (Fig 4C), which was absent in time-shuffled data (Fig 4D). While the presented model may thus provide potential insight into the underpinnings of the experimental observations, it needs to be stressed that also other mechanisms may underlie these observations which future research may attend to.

## Discussion

Accumulating evidence suggests temporal and spatial correlations to reflect integration properties of neural circuits in time and space. Using high-density mapping across full days of

recording and different cortical regions, we here showed that these indices fluctuate and exhibit systematic co-variation. We observed that temporal correlations align with functional hierarchy in cortex, and that spatio-temporal correlations decline under AED action and break down during SWS. The results may thus have implications for characterizing the brain's changing information processing capabilities.

Why do spatial and temporal correlations exhibit such strong co-variation? A large body of research suggests that cortical dynamics may reside in the vicinity of a phase transition between vanishing and collective activation, which can be described as a critical branching process [19,24,25,28–37]. The hallmark of the critical state are the long-range correlations which decay algebraically there [38]. This means that autocorrelation and cross-correlation functions, as studied here, will decay slowest there [20,39]. Thus, when dynamics fluctuate in the vicinity of a critical point, spatial and temporal correlations may naturally co-vary, i.e., increase or decrease when dynamics are closer or further away from this point, respectively, as evident in the companion neural network model. A systematic decline of spatio-temporal correlations, as seen in model and data, is expected when dynamics are driven towards the subcritical regime by AED [39]. It has been shown in rodent models that the breakdown of TC and SC during SWS may be caused by the ability of neurons to go offline during SWS, thus disrupting any auto- and cross-correlation structures in neuron activity [40,41]. This disruption of dynamics may thus similarly indicate a shift away from the critical point and potentially provide an explanation why no differences in TC and SC for different AED loads were observable during SWS in our data.

It is important to note that, while the model studied here may indicate one potential mechanism with regards to the experimental observations, this mechanism may certainly not be the only one, and other factors and mechanisms may be at play. For example, other models, e.g. based on coupled Kuramoto oscillators [42,43], may also exhibit a critical point and potentially similar behaviour. Another interpretation of our data may posit that each of the cortical regions and dynamics under AED loads could potentially still be perfectly poised at criticality but would have their own critical exponent and thus their own scaling function for autocorrelation coefficients, resulting in different TC values. Another potential explanation for the covariation of TC and SC could be given by slow changes of an external drive. However, this might seem unlikely based on investigations over 48 hours and the systematic changes observed under AED load or with cortical hierarchy. In particular the ability to tune a system in a way that leads to a collapse of signatures of criticality has been argued to be a strong indicator for critical dynamics emerging from network interactions in the unperturbed state [44,45]. In our analysis, the increased AED loads can be regarded as tuning mechanisms of the system, i.e., decreasing the network connectivity, which lead to decreased STC as measures of the distance to criticality.

We here report to our knowledge for the first time in humans that temporal correlations within highly resolved electrophysical signals increase along the functional hierarchy. This is in agreement with observations in non-human primates using single-unit activity [6] and results from a fMRI study in humans [12]. It is also in alignment with other recent studies showing that more complex tasks, often associated with regions higher in the functional hierarchy, might profit from longer temporal correlations [46,47]. Together, these findings may thus indicate that long temporal correlations could be beneficial for information integration especially in regions higher within the functional hierarchy. The neural network model studied here may exhibit such a gradient of temporal correlations by tuning only one parameter in contrast to previous models explaining such gradients [1]. Specifically, when tuning the model closer to criticality, dynamics exhibited longer spatio-temporal correlations and thus longer memory within the signal [38]. Extended memory (or also self-affinity) in a signal is also often quantified by the Hurst exponent which is 0.5 for uncorrelated signals and grows up to 1 for

correlated, self-affine signals. Our model closely matched this expected behavior as it showed Hurst exponents close to 0.5 for simulations far from criticality ($\lambda \neq 1$) and Hurst exponents around 1 at the critical value of the absolute value of the largest eigenvalue ($\lambda = 1$). Our model, however, consists only of one network, and may thus only mirror one cortical region at a time. Therefore, it does not account for region-to-region interactions which, besides other effects like neural transmission speeds, might play an important role leading to the correlation structure observed in the data. Hence, further research is warranted to validate the mechanism leading to the spatial and temporal correlation gradients we observed.

Long spatio-temporal correlations along with other capabilities [21,31,35,48–51] may provide some benefits for information processing as information can be stored for longer periods and inputs from more sites can be integrated. Conversely, when spatio-temporal correlations decline, as under AED or during SWS, this may potentially impair information processing. Impairments of cognition and perception are widely observed side effects under AED treatment [52–56] and can be observed for neurological disorders as well [57,58]. Spatio-temporal correlations may therefore potentially serve as biomarkers for these deficits in future research.

Even though, we accounted for many confounding factors, including AED load, SWS, seizures and subclinical events, removal of all channels with interictal epileptiform discharges, it needs to be noted that the data still stem from patients with epilepsy which may limit transfer of these concepts to healthy subjects. For instance, cortical dynamics may potentially take multiple days to relax back to "normal" after surgery for electrode implantation. As the AED load often correlates with the time after surgery this could potentially be another, at least partial contributor to the observed changes in STC under changing levels of AED. In our data, 17 out of 23 patients had their higher medication day closer to the surgery date than the lower medication day. A sub-analysis of the six patients with the low AED day closer to the surgery, however, did not show a significant opposite effect in the other direction, which might have been an indication of the surgery effect. Thus, while the current data does not suggest time to surgery to be a main driving factor, further research is needed to better delineate the effects of AED load and relaxation of cortical dynamics after surgery. Furthermore, these effects should also not affect changes of STC during SWS and along the cortical hierarchy, as these were compared within patients and did not have a timely delay to surgery.

Furthermore, the sleep staging used in this manuscript is only able to detect SWS. Even though, more elaborate algorithms which are also able to detect other sleep stages from iEEG are in principle available they are not applicable to our data set due to the sampling frequency [59] or data set duration [60]. While our research highlights the importance to study temporal and spatial correlations over extended periods of time to capture their variability, the sampling across cortical areas is naturally limited by electrode positioning determined by the clinical need. This leads to a relative under-sampling for example of the ACC in comparison to other regions. This sparse spatial sampling also limited us in calculating spatial correlations for individual regions along the hierarchy. Empirical evidence indicative of a strong co-variation between spatial and temporal correlations along with conceptual insights from our model, however, strongly suggest that also spatial correlations should follow a gradient along the functional hierarchy, similar to temporal correlations. Future research should thus address further validation of the spatial structure of temporal and spatial correlations, especially in such parts of the cortex.

## Methods

### Ethics statement

Multi-day invasive electroencephalographic (iEEG) recordings of 23 patients with epilepsy undergoing presurgical evaluation at the Epilepsy Center of the University Hospital of

Freiburg, Germany, were analysed (12 female, age 28±13 years, mean ± standard deviation). The data set was made available in the *Epilepsiae* database, their use for research was approved by the ethics committee of the University of Freiburg and written informed consent that the clinical data might be used and published for research purposes was given by all patients [61]. The study was approved by the local institutional review board (EK 92022019). Detailed data on the included patients can be found in the S2 Table.

## Preprocessing of invasive electroencephalography data

Data from two full days (midnight to midnight each), the day with highest and lowest anti-epileptic drug (AED) loads, were analysed in each patient. The day with highest and lowest AED loads were determined by the summed prescribed drug dosages normalized by their individual defined daily dosages across drugs. If more than one day had the same AED load, highest and lowest AED days were chosen so that the time between days was maximized. Data from each day was processed in consecutive 1 h segments. First, power line noise and its first harmonic were removed using a notch filter at 50 and 100 Hz. Second, iEEG data, which was sampled at either 256, 512 or 1024 Hz, was down-sampled to the common frequency 256 Hz by applying an anti-aliasing filter and then decimating the signal. After visual examination occasional 1 h data segments with additional frequency-restricted artefacts were removed from further analysis. All epileptic seizures, including a 10-min preictal and 10-min postictal period, were excluded for the purpose of this study. Furthermore, all subclinical seizures, including 2 min before and after, as well as all channels containing interictal discharges as labelled in the Epilepsiae database, were excluded from the analysis [61].

## Assignment of electrodes to brain regions

The number of electrodes varied between patients and included both surface and depth electrodes. Electrode placement was solely determined by clinical considerations. Individual brain surfaces were constructed from MRI images and warped onto a common standard MNI-152 template using the software package *AFNI* [62]. As a result, electrode positions were assigned to a certain region if they fell within 9.5 mm of it (Euclidean distance), which is a trade-off between positional accuracy and maximising the number of electrodes for each region. Generally, the regions are far apart allowing for such a coarse assignment and an observation of posterior to frontal changes of our measures. Electrodes were excluded when their positions were not differentiable. Five human brain areas along the visual pathway were chosen, functionally corresponding to the five regions investigated in non-human primates in [6]. Hence, we adopted the names medial temporal (MT) and lateral intraparietal (LIP) area in visual cortex, lateral prefrontal (LPFC), orbitofrontal (OFC) and anterior cingulate cortex (ACC). The function associated with LIP in non-human primates can be more likely associated with the medial part of the intra-parietal sulcus in humans [63]. Therefore, we investigated this brain area but kept the name LIP for comparability to [6]. All regions were drawn as set of regions constructed in [64]. In Fig 1 all regions except ACC can be seen. The full list of the sets for each region can be found in the S1 Table.

## Quantification of temporal correlations

Power fluctuations in the high $\gamma$-band have been shown to provide a local, spatio-temporal estimate of population spike rate variations near an electrode [14–18]. We therefore evaluated the fluctuations in the median power between 56–96 Hz and, for simplicity, refer to this frequency band as broadband $\gamma$-power. Following previous work, for each iEEG channel, the time series of broadband $\gamma$-power fluctuations were obtained by calculating the power every

125 ms (Welch's method, Hanning window; [5,19]). We observed the median broadband $\gamma$-power to be approximately log-normally distributed (see S1 Fig) and therefore applied the logarithm with base 10 to the time series (see middle panel in Fig 1 for a typical time series).

Next, autocorrelation functions were calculated from consecutive, 75% overlapping 2-min data segments of these individual-channel power time series (see top right panel in Fig 1). Following previous work, temporal correlations (TC) were defined as the time lag when the autocorrelation function dropped for the first time below half of the difference between the value at the first lag and the baseline (red dashed line in Fig 1). The baseline was defined as the median value between 40 and 60 sec when autocorrelation values had typically settled to a minimum. Results do not depend on the exact choice where the baseline was defined, as one can see in the S4 Fig for a baseline defined as median between 15–30 seconds or a baseline set to 0.

Additionally, note that the minimal value of the TC measure is one time lag, i.e., 0.125 seconds, which leads to a non-zero TC measure even in surrogate data (defined below).

## Quantification of spatial correlations

Spatial correlations (SC) were calculated on the same broadband $\gamma$-power time series as temporal correlations. Specifically, for all possible pairs of channels, the Pearson cross-correlation was calculated for each consecutive, non-overlapping 2-min segments. Correlation values from channel pairs were averaged according to their Euclidean distance with a bin size of 1 mm to get a distance dependent cross-correlation function. SC were then defined as the average cross-correlation within the 7 to 79 mm interval (grey shaded area within the cross-correlation panels in Fig 1). The interval was chosen as most of the patients had electrodes within these distances (Fig 2F). However, results did not change when other distances were used, see S5 Fig. As this is a direct measure of the cross-correlation function it is expected to be zero in surrogate (time-shuffled) data, as can be seen in Fig 2F.

## Surrogate data

As additional controls, SC and TC were also calculated in the same way from randomly time-shuffled power time series (see lower panels in Fig 1). Specifically, after extracting the broadband $\gamma$-power time series, this power time series was then randomly shuffled within each 2-min segment. As the time series shuffling only happens after the power spectral density is computed the power distribution in the data is preserved but the temporal correlations in the data are eliminated. This was done for each electrode separately and, by consequence, eliminated spatial correlations as well. Using these time-shuffled surrogate data, TC and SC were calculated. Comparison of our data results with these surrogate data allowed to rule out that our findings simply stem from fluctuations in the power but were indeed caused by changes in the correlations in space and time.

## Staging of vigilance states

To evaluate spatial and temporal correlations (STC) as a function of vigilance state, specifically during and outside of slow-wave sleep (SWS), we calculated a vigilance index following previous work [23] which is defined on 30-sec windows as the band power ratio

$$\frac{\Theta + \delta}{\alpha + \beta_{\mathrm{high}} + \mathrm{spindle}}. \tag{1}$$

Segments were classified as SWS whenever the respective vigilance index was one standard deviation above the mean vigilance index for the given full day [23]. Full 24-hour days were

scored individually for each patient to decrease the effects of possible multi-day rhythms and potential impact under changing AED levels. A 2-min segment was finally classified as SWS only if all 30-sec segments in it were classified as SWS and as nonSWS otherwise.

## Neural network model

We studied STC as a function of network state in a parsimonious neural network model. In the model $N = 1600$ neurons were distributed on an equidistant 2-dimensional grid with periodic boundary conditions. The weights of an all-to-all connected adjacency matrix were first assigned random strengths from a uniform distribution which then were adjusted by a Gaussian profile

$$e^{-\frac{r_{ij}^2}{2\sigma^2}},\tag{2}$$

with $r_{ij}$ being the Euclidian distance between neuron $i$ and neuron $j$ and $\sigma$ scaling the width of the profile similar to [65,66]. Self-connections were omitted and for all calculations shown $\sigma$ was set to $\sigma = 4$ as a trade-off between a fully connected and a not connected network as in Ref. [66]. A total of $\alpha = 20\%$ of the neurons was randomly set to be inhibitory, i.e., their weights are multiplied by $-1$ and the remaining 80% of neurons were excitatory, i.e., they had positive outgoing connections. To model the influence of AED the weights of inhibitory and excitatory neurons in the weight matrix $\underline{\underline{w}}$ can be separately scaled by a factor $f_{\text{inh}}$ and $f_{\text{exc}}$, respectively. The unperturbed network is given for $f_{\text{inh}} = f_{\text{exc}} = 1$ and increasing only $f_{\text{inh}}$ leads to stronger inhibition whereas decreasing $f_{\text{exc}}$ leads to decreased excitable output of the excitatory neurons.

Each neuron $i$ can be in either of two states $s_i(t) \in \{0,1\}$ at any timepoint $t$. Here, $s_i(t) = 0$ corresponds to the neuron not firing and $s_i(t) = 1$ to the neuron firing. The firing probability in the next time step $p_i(t+1)$ is determined by the sum over all inputs and takes the form

$$p_i(t+1) = \begin{cases} 0 \text{ if } \sum_j w_{ij}s_j(t) \leq 0 \\ 1 \text{ if } \sum_j w_{ij}s_j(t) \geq 1 \\ \sum_j w_{ij}s_j(t) \text{ otherwise} \end{cases},\tag{3}$$

with $w_{ij}$ being the connection strength of neuron $i$ and $j$. To study the dynamics with some background activity, every time step one random neuron is activated, and each simulation is run until a maximal number of time steps, $\max_t$, is reached (results are independent of the choice of $\max_t$ as long as the network simulation is running long enough to get a robust estimation of the auto-correlation function). Additionally, similar results can be obtained for lower background activation rates.

Generally, for this type of model macroscopic dynamics are governed by the absolute value of the largest eigenvalue $\lambda$ of the weight matrix $\underline{\underline{w}}$ [24,25]. If $\lambda > 1$ the dynamics are generally supercritical and the average activity $S(t) = N^{-1}\Sigma_i s_i(t)$ approaches 1, i.e., excites the whole network. If $\lambda < 1$ dynamics are subcritical and $S(t)$ approaches 0. When reporting $\lambda$ in this paper we always refer to the case unperturbed by AED action, i.e., $f_{\text{exc}} = f_{\text{inh}} = 1$. To model dynamics in different cortical areas with varying distance to criticality, the largest absolute eigenvalue was chosen from $\lambda \in \{0.9, 0.925, 0.95, 0.975, 1\}$ and for each $10^4$ simulations were run ($\lambda$ can be adjusted by multiplying every matrix entry by a constant factor). To calculate SC, activity was

monitored for 40 random neurons to calculate their cross-correlation in each simulation. SC were then determined as the mean value of the cross-correlation function over a distance between 1 and 10 arbitrary units. Results did not depend on the number of neurons being monitored or the exact choice of evaluated distance between them. TC were calculated from the average activity trace of each simulation.

## Statistical analysis

For comparisons between different states, like nonSWS and SWS, the data was balanced by under-sampling the majority group to ensure that the same number of data segments were included in both groups. This was done to rule out the possibility that the difference in number of segments was (partly) causing the observed effects. Results were, however, robust when this balancing was not performed. At the group level, non-parametric paired Wilcoxon tests were used to test for significant effects between different states. Individual slopes obtained by applying a linear least squares fit were calculated on the patient level for the hierarchy analysis.

## Supporting information

**S1 Table. Abbreviations for cortical regions along the visual pathway in non-human primates as in [6] and their counterparts in humans defined from the parcellation in [64].**
(XLSX)

**S2 Table. Meta data of the patients from the EPILEPSIAE database [61].** Drug loads are given in fractions of the defined daily dosage, see S3 Table for full names of the drugs. Total number of electrodes are given with the number of electrodes showing interictal epileptiform discharges (IED) in parenthesis.
(XLSX)

**S3 Table. Names of drugs, their abbreviations as used in S2 Table and their defined daily dose.**
(XLSX)

**S1 Fig. Distribution of median broadband $\gamma$-power for one exemplary patient.** Left: Histogram of the median broadband $\gamma$-power. Right: Same data transformed by applying the logarithm with base 10. Black lines in both panels show a fitted log-normal distribution.
(TIF)

**S2 Fig. Temporal correlations for different standard bands with exclusion of subclinical seizures and interictal epileptiform discharges.** In each side-by-side plot only one state is changed, either the drug load or the sleep stage.
(TIF)

**S3 Fig. Spatial correlations for different standard bands with exclusion of subclinical seizures and interictal epileptiform discharges.** In each side-by-side plot only one state is changed, either the drug load or the sleep stage.
(TIF)

**S4 Fig.** TC calculated from broadband $\gamma$-power for A the baseline defined as the median value between 15 and 30 seconds and B the baseline set to 0 for the calculation of the half maximum value. Results are quantitively the same as for the definition used in the main manuscript (baseline at the median value between 40 and 60 seconds, compare panel E in S2 Fig).
(TIF)

**S5 Fig.** SC calculated from broadband $\gamma$-power defined as the area under the cross-correlation function between A 9 and 50 mm and B 50 and 79 mm. Results are quantitively the same as for the definition used in the main manuscript (area under the cross-correlation function between 9 and 79 mm, compare panel E in S3 Fig).
(TIF)

**S6 Fig. Covariation of SC and TC for first 8 patients. (m.p. = machine precision).**
(TIF)

**S7 Fig. Covariation of SC and TC for second 8 patients.** (m.p = machine precision).
(TIF)

**S8 Fig. Covariation of SC and TC for last 7 patients.**
(TIF)

**S9 Fig. Measures of correlation in the model for different absolute values of the largest eigenvalue $\lambda$.** All model parameters are as in the main text of the manuscript (compare Fig 4 in the main text). A Spatial correlations (SC), as defined in the main text of the manuscript (area under the cross-correlation function (AUC)), increase beyond the critical value of $\lambda = 1$. B Temporal correlations (TC), as defined in the main text of the manuscript, peak at $\lambda = 1$ indicative of a critical point of the model at $\lambda = 1$. C Defining SC similar to TC (i.e., first distance of the correlation function to fall below half the value between the value at the first distance and the baseline) shows SC to peak at the critical value of $\lambda = 1$. This is only visible in the average (shown here) due to the model being restricted in spatial resolution (only 20 different distances) and single model simulations being noisy leading to non-smooth cross-correlation functions (not shown). Importantly, calculating SC this way is only possible in the model as short interelectrode distances are generally missing in EEG data, which are, however, essential for this quantification (compare Fig 2 in the main text). D The Hurst exponent, as calculated in Hardstone et al., 2014 [26], peaks at $\lambda = 1$. This indicates higher self-affinity of the signal the closer it is initiate to criticality. Such self-affinity cannot be observed in the surrogate (time-shuffled) data (grey bars). Hurst exponents were calculated on scales from 100 to 1000 time steps (model simulations were 2000 time steps long).
(TIF)

## Author Contributions

**Conceptualization:** Christian Meisel.

**Formal analysis:** Paul Manuel Müller.

**Methodology:** Christian Meisel.

**Supervision:** Christian Meisel.

**Visualization:** Paul Manuel Müller.

**Writing – original draft:** Paul Manuel Müller, Christian Meisel.

**Writing – review & editing:** Paul Manuel Müller, Christian Meisel.

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
