## [Decision Letter · Decision Letter 0]

3 Dec 2022

Dear Dr. Meisel,

Thank you very much for submitting your manuscript "Spatial and temporal correlations in human cortex are inherently linked and predicted by functional hierarchy, vigilance state as well as antiepileptic drug load" for consideration at PLOS Computational Biology.

As with all papers reviewed by the journal, your manuscript was reviewed by members of the editorial board and by several independent reviewers. In light of the reviews (below this email), we would like to invite the resubmission of a significantly-revised version that takes into account the reviewers' comments.

We cannot make any decision about publication until we have seen the revised manuscript and your response to the reviewers' comments. Your revised manuscript is also likely to be sent to reviewers for further evaluation.

Sincerely,

Peter Neal Taylor

Academic Editor

PLOS Computational Biology

Lyle Graham

Section Editor

PLOS Computational Biology

Reviewer's Responses to Questions

**Comments to the Authors:**

Reviewer #1: In this manuscript by Muller and Meisel, the authors investigate spatial and temporal correlations in data from human patients. They conclude that spatial and temporal correlations are significantly linked, that they are greatest in cortical regions that are highest in the functional hierarchy and that they decline under anti-epileptic drugs. The findings in the data are helpfully replicated by a simple computational model that captures features of criticality. This model shows that spatial and temporal correlations are maximal near the critical point and that they will decline when the excitatory/inhibitory balance is disrupted, as would be expected with anti-epileptic drugs.

It has been known for nearly two decades that long-range temporal correlations may be connected to criticality in the human brain (Linkenkaer-Hansen et al., 2001; the authors cite this). Likewise, there has been work in monkeys, which the authors also cite, showing that temporal correlations are greatest in cortical areas that are highest in the hierarchy (Murray et al., 2014). Similarly, the role of spatial correlations in humans has been previously examined and linked to criticality (Expert et al., 2010; the authors cite this). While these findings have been reported, the connections between them in human subjects has been lacking. The appeal of the manuscript by Muller and Meisel is that all of these disparate results are now brought together in one population of humans and examined for their interrelationships. This made the manuscript very satisfying for me to read and scientifically verified some of the relationships that had been hypothesized by others. Now we know that spatial and temporal correlations are related to each other in humans; previously this had only been suspected. Now we know that the correlations climb within the human cortical hierarchy; we know that anti-epileptic drugs cause them to decline. The paper is clearly written for the most part (see a few suggestions for improvement below) and nicely presented with a wealth of data.

In what follows, I offer suggestions for the authors’ consideration for possible improvement.

Line 198 and thereabouts: It might be helpful here if you could explain a little more about the model, even though you give references. Specifically, readers may want to know if there are both excitatory and inhibitory neurons and if there are feedback loops and any plasticity mechanisms that could enhance long-range temporal correlations (LRTCs). A sentence or two could tide them over until they dig it up in the supplementary information.

Figure 2: Are the colors switched here between plots E, F compared to G, H? I thought orange was high AED in E, F. Yet it is low AED in the histogram plots G, H?

Figure 3: Could you also look at local spatial correlations within the hierarchy (for example, within OFC compared to within MT)? If so, did those increase also? I realize that such an analysis would have to examine data within a cortical region, and you might not have enough closely-spaced electrodes for that. If that is the case, just say so.

Around line 203: Here you are talking about correlations in the model as it was tuned from the subcritical phase up to the critical point. I noticed that you did not tune the model beyond the critical point into the supercritical region: lambda ranged from {0.9, 0.925, 0.95,0.975, 1}. Did spatio-temporal correlations (STCs) decline again when lambda was increased beyond 1 (supercritical)? Can you show us the other (supercritical) side of the histograms in Figure 4 E, F? If STCs continue to be high with correlations, you could consider switching to covariance (as opposed to just correlation), which should show a decline in the supercritical region (see the Ising model in Beggs and Timme, 2012). The basic idea is that there are times when the correlation will be high but the covariance will be low.

In many places you mention temporal correlations, TCs, but you do not use a measure that has been widely adopted in the human literature: the Hurst exponent. Since many other human data papers use it (e.g., Hardstone et al., 2012; Ihlen et al., 2012) it may be good for readers to know how your TCs map onto typical values of the Hurst exponent. I would imagine that it is difficult for your model to produce Hurst exponents much larger than 0.5, as would be expected in a random walk process. As you probably know, this was one of the main shortcomings of the branching model as pointed out by (Poil et al., 2008; which you cite). It is often difficult to get larger values of the Hurst exponent without adding in feedback loops or some type of plasticity in the network. If you do not have H > 0.5, that is fine. But you should say so to clearly delineate the boundaries of your model.

You mention anti-epileptic drugs (AED) many times and show that they all produce roughly the same effects in terms of causing correlations to decline. I also noticed that you have a table of them in the supplementary material. It might be helpful to include a few sentences for readers who do not specialize in this area of pharmacology, addressing a few questions: Were all AED of the same types or were they different? If different, can you briefly explain how? For example, were they excitatory blockers or inhibitory agonists? Did your model take this into account? Could they have statistically different effects? Did you examine this? I am fine with you summarizing their effects. I just felt unsure if they were all doing roughly the same thing at the mechanistic level, given that there were so many of them, and at widely different dosages. A summary would be helpful to the reader.

In lines 261-265 in the discussion, you mention the possible confound of an external driving source causing “apparent criticality” in your human data. It is good that you bring this up, as this has been relatively under-appreciated in the criticality community at times. Some very recent work has shown that you can distinguish between criticality caused by external drive from criticality caused by emergent dynamics (Mariani et al., 2022). Relevant for your case, if blockers of synaptic transmission cause signatures of criticality to fade or disappear (as in Meisel, 2020, PNAS, and this paper), then you can largely rule out the hypothesis that the signatures of criticality are caused by a random external drive (Beggs, 2022). Rather, they emerge through the collective interactions of neurons and cortical regions.

References

Mariani, Benedetta, Giorgio Nicoletti, Marta Bisio, Marta Maschietto, Stefano Vassanelli, and Samir Suweis. "Disentangling the critical signatures of neural activity." Scientific Reports 12 (2022): 10770.

Beggs, J. "Addressing skepticism of the critical brain hypothesis." Frontiers in computational neuroscience 16 (2022).

Reviewer #2: The manuscript presents a highly detailed analysis of temporal and spatial correlations in intracranial EEG data. In general, the methods are rigorous and the manuscript well written. Particular strengths include using 48 hours of recording per subject and analyzing the effect of state of vigilance and medication load. The only significant weakness I noted was the lack of addressing the impact of the number of days since surgery, as the brain can take days to weeks to return to normal after surgery. Drug load is also correlated with time since surgery, as clinicians may initially taper medications down to increase seizure likelihood and then later increase them once enough seizures are recorded.

Minor comments follow:

Results: define gamma frequency range when gamma-power is first mentioned

How do shuffled data with same gamma power

“As a result, electrode positions were assigned to a certain region if they fell within 9.5 mm of it” A threshold of 9.5 mm seems a bit too large. Can you provide any data or better justification?

“which leads to a non-zero TC measure even in surrogate data” Please define the surrogate data more clearly. How was it created? What exactly did you do with it? This could be an additional strength to the manuscript, but as of now, surrogate data gets a very minimal mention.

“A 2-min segment was finally classified as SWS, if all 30-sec segments in it were classified as SWS.” What was done for the 2-minute segments that only partially contained SWS? Were they included in the non-SWS group or where they ignored?

**Have the authors made all data and (if applicable) computational code underlying the findings in their manuscript fully available?**

Reviewer #1: Yes

Reviewer #2: Yes

PLOS authors have the option to publish the peer review history of their article (what does this mean?). If published, this will include your full peer review and any attached files.

Reviewer #1: **Yes: **John M. Beggs

Reviewer #2: No
---

## [Decision Letter · Decision Letter 1]

23 Jan 2023

Dear Dr. Meisel,

Thank you very much for submitting your manuscript "Spatial and temporal correlations in human cortex are inherently linked and predicted by functional hierarchy, vigilance state as well as antiepileptic drug load" for consideration at PLOS Computational Biology. As with all papers reviewed by the journal, your manuscript was reviewed by members of the editorial board and by several independent reviewers. The reviewers appreciated the attention to an important topic. Based on the reviews, we are likely to accept this manuscript for publication, providing that you modify the manuscript according to the review recommendations.

Sincerely,

Peter Neal Taylor

Academic Editor

PLOS Computational Biology

Lyle Graham

Section Editor

PLOS Computational Biology

Reviewer's Responses to Questions

**Comments to the Authors:**

Reviewer #1: I think the authors have done a good job of addressing my questions. It is a solid paper, providing important information about humans that we did not know previously.

Reviewer #2: I appreciate the authors attention to the previous comments and their work to improve the manuscript. The manuscript will be an important contribution to this field. I have one remaining concern regarding how the surrogate data was generated. First, I'd like to point out that I greatly appreciate the authors use of surrogate data, which strengthens the paper significantly. I also appreciate the further explanation of the surrogate data generation method which has been added. However, the explanation of the method brings up an essential question.

It sounds like you have just randomly permuted the time series data. This will not maintain the power distribution as you claim. Reordering the time series points will change the power distribution, which you can easily verify by computing power spectrums before and after the random permutation. Also consider two extreme examples: ordering the points in descending order versus ordering the points based on every other point being descending from the max value or ascending from the min value. One will include only low frequencies, and one will include high frequencies. It is unclear whether your method is valid but the explanation is lacking or whether your chosen method has fundamental flaws.

Standard methods exist for creating surrogate data with a power spectrum that matches given data. On option is IAAFT, Schreiber & Schmitz (1996) [doi:10.1103/PhysRevLett.77.635]. A more robust method is the autoregression/all-pole filter approach used in, e.g., Roehri et al (2017) [doi:10.1371/journal.pone.0174702] and Gliske et al. (2020) [doi:10.1088/1741-2552/abb89b]. I'd recommend using the autoregression/all-pole filter approach.

**Have the authors made all data and (if applicable) computational code underlying the findings in their manuscript fully available?**

Reviewer #1: Yes

Reviewer #2: Yes

PLOS authors have the option to publish the peer review history of their article (what does this mean?). If published, this will include your full peer review and any attached files.

Reviewer #1: **Yes: **John M. Beggs

Reviewer #2: No

Figure Files:

Data Requirements:

Reproducibility:

References:

---

## [Editor Report · Decision Letter 2]

3 Feb 2023

Dear Dr. Meisel,

We are pleased to inform you that your manuscript 'Spatial and temporal correlations in human cortex are inherently linked and predicted by functional hierarchy, vigilance state as well as antiepileptic drug load' has been provisionally accepted for publication in PLOS Computational Biology.

Best regards,

Peter Neal Taylor

Academic Editor

PLOS Computational Biology

Lyle Graham

Section Editor

PLOS Computational Biology

---

## [Editor Report · Acceptance letter]

27 Feb 2023

PCOMPBIOL-D-22-01434R2 

Spatial and temporal correlations in human cortex are inherently linked and predicted by functional hierarchy, vigilance state as well as antiepileptic drug load

Dear Dr Meisel,

I am pleased to inform you that your manuscript has been formally accepted for publication in PLOS Computational Biology. Your manuscript is now with our production department and you will be notified of the publication date in due course.

With kind regards,

Timea Kemeri-Szekernyes
